# Comparative Genomics of *Lotus japonicus* Reveals Insights into Proanthocyanidin Accumulation and Abiotic Stress Response

**DOI:** 10.3390/plants13081151

**Published:** 2024-04-20

**Authors:** Zhanmin Sun, Ziyang Liu, Manqing Zhi, Qifan Ran, Wenbo Xue, Yixiong Tang, Yanmin Wu

**Affiliations:** 1Biotechnology Research Institute, Chinese Academy of Agricultural Sciences, Beijing 100081, China; liuzy17@lzu.edu.cn (Z.L.); zmq990305@163.com (M.Z.); tangyixiong@caas.cn (Y.T.); wuyanmin@caas.cn (Y.W.); 2Chongqing Academy of Animal Sciences, Chongqing 402460, China; ranqifan@outlook.com; 3BGI Genomics, Shenzhen 518085, China; xuewenbo@genomics.cn

**Keywords:** *Lotus japonicus*, proanthocyanidins, RNA-seq, abiotic stress, comparative genomics

## Abstract

*Lotus japonicus*, is an important perennial model legume, has been widely used for studying biological processes such as symbiotic nitrogen fixation, proanthocyanidin (PA) biosynthesis, and abiotic stress response. High-quality *L. japonicus* genomes have been reported recently; however, the genetic basis of genes associated with specific characters including proanthocyanidin distribution in most tissues and tolerance to stress has not been systematically explored yet. Here, based on our previous high-quality *L. japonicus* genome assembly and annotation, we compared the *L. japonicus* MG-20 genome with those of other legume species. We revealed the expansive and specific gene families enriched in secondary metabolite biosynthesis and the detection of external stimuli. We suggested that increased copy numbers and transcription of PA-related genes contribute to PA accumulation in the stem, petiole, flower, pod, and seed coat of *L. japonicus*. Meanwhile, According to shared and unique transcription factors responding to five abiotic stresses, we revealed that MYB and AP2/ERF play more crucial roles in abiotic stresses. Our study provides new insights into the key agricultural traits of *L. japonicus* including PA biosynthesis and response to abiotic stress. This may provide valuable gene resources for legume forage abiotic stress resistance and nutrient improvement.

## 1. Introduction

Legumes, as the second most crucial plant in the world, contribute a large amount of protein and nutrients as well as provide a vital ecosystem source of reduced nitrogen through symbiotic nitrogen fixation. *Lotus japonicus* is a perennial model legume forage closely related to the cultivated species *L. corniculatus*. Nowadays, because of the availability of high-quality genome assemblies [1,2,3], easy genetic transformation, and availability of substantial mutant resources, *L. japonicus* has been extensively used for studying molecular mechanisms related to plant–microbial symbiosis, secondary metabolism biosynthesis, particularly proanthocyanidins (PAs), and responses to biotic and abiotic stresses.

Firstly, *L. japonicus* is proposed as an ideal model system for studying regulatory mechanisms of proanthocyanidin (PA) biosynthesis. PAs or condensed tannins (CTs) are polymeric flavonoids contributing quality traits to crucial forage legumes due to preventing pasture bloat and decreasing ruminant methane emissions. Unfortunately, the most crucial pasture legumes such as alfalfa (*Medicago sativa*) and white clover (*Trifolium repens*) cannot accumulate PAs in the leaf and stem tissues, whereas, *L. corniculatus* contains moderate PA levels in the leaves, stems, and other tissues. As a model legume, *L. japonicus*, which possesses PA in most organs excluding leaves, can provide an important source of information for increasing PA levels in alfalfa or white clover leaves and stems through genetic engineering and breeding. Although the genome of *L. japonicus* has been published, comparative genomics and evolutionary analyses of the genes associated with PA biosynthesis are lacking.

Meanwhile, abiotic stresses (including drought, salinity, heat, cold, and heavy metal stresses AlCl_3_) are considered major limiting factors that qualitatively and quantitatively affect crop production [4]. In fact, numerous *Lotus* species and cultivars exhibit wide and different degrees of tolerance. Almost all *Lotus* species and cultivars are more tolerant of abiotic stress than *Medicago* (lucerne) or *Trifolium* (trefoils) [5].

In this study, based on our previous de novo assembled and annotated *L. japonicus* MG-20 genome [3], we compared the *L. japonicus* genome with those of other legume species, and revealed the expansive and specific gene families enriched in the biosynthesis of secondary metabolites and detection of external stimuli. Expansion of the first key enzyme phenylalanine ammonia lyase (*PAL*) and genes in the late biosynthesis pathway that encode dihydroflavonol 4-reductase (*DFR*) and transparent testa 2 (*TT2*), and transcription expression of genes such as flavonoid 3′,5′-hydroxylase (*F3′5′H*), dihydroflavonol 4-reductase (*DFR*), leucoanthocyanidin dioxygenase (*ANS*), and anthocyanidin reductase (*ANR*) in most tissues mainly contribute to PA accumulation in the stems, petioles, flowers, pods, and seed coat of *L. japonicus*. After studying the shared and unique transcription factors (TFs) that respond to five abiotic stresses, we suggested that MYB and AP2/ERF play more prominent roles in abiotic stress responses. This may provide valuable gene resources for legume forage abiotic stress resistance and nutrient improvement.

## 2. Results

### 2.1. Comparison of Genome Changes in Sister Legume Species

In our previous study, we sequenced, assembled, and annotated the *L. japonicus* MG-20 genome [3]. Based on the aforementioned works, we performed an evolutionary analysis according to the alignment of 592 single-copy gene families shared by a total of eight legume species (*L. japonicus*, *M. truncatula*, *G. max*, *T. pretense*, *P. vulgaris*, *C. arietinum*, *V. unguiculata*, and *Lupinus angustifolius*) and a nonlegume species (*A. thaliana*). The constructed phylogenetic tree not only showed close relationships among *L. japonicus*, *M. truncatula*, *T. pretense*, and *C. arietinum* but also reflected the divergence between *L. japonicus* and the common ancestor of *M. truncatula*, *T. pretense*, and *C. arietinum*, which occurred approximately 48.6 MYA (Figure 1A).

The 38,684 gene sets of *L. japonicus* were compared with 292,810 gene models from the seven sequenced legumes and one nonlegume species (*A. thaliana*), and 36,258 orthogroups and 16,416 families (orthologous groups) were identified using the software OrthoFinder ver.2.3.8 (Appendix A). Among them, 7215 orthologs contained only a single *L. japonicus* gene, suggestive of simple orthology (Appendix A and Figure 1B). On the other hand, a total of 10,902 families (orthologous groups) were shared in all nine species, and 773 gene families were composed of only *L. japonicus* proteins (Appendix A and Figure 1C). Gene Ontology (GO) and Kyoto Encyclopedia of Genes and Genomes (KEGG) enrichment analysis of *L. japonicus*-specific genes revealed that these gene families were significantly enriched in genes related to secondary metabolite biosynthesis, detection of external stimuli, circadian rhythm, selenocompound metabolism, and regulation of macromolecule biosynthesis (Figure 2A,B). 

Furthermore, we identified the expanded or contracted gene families. Our results revealed that 112 families comprising 1126 genes exhibited significant expansion in the *L. japonicus* genome (Appendix A). These genes were functionally annotated. Functional annotation demonstrated that the genes were mainly enriched in functional categories, namely secondary metabolite biosynthesis, plant–pathogen interaction, polysaccharide metabolism, and response to abiotic stimuli (Figure 2C,D and Appendix A). Furthermore, 18 families comprising 176 genes exhibited significant contraction in the *L. japonicus* genome (Appendix A), and these genes were significantly enriched in functions related to phosphorus metabolism and S-adenosylmethionine biosynthesis/metabolism (Figure 2E). 

Synteny block analysis is generally used to study chromosome evolution among related species. Here, we analyzed the aligned protein sequences of *L. japonicus* in comparison with those of *M. truncatula*. The synteny results indicated the presence of shared 20,024 gene pairs in 833 synteny blocks (Appendix A). *L. japonicus* chromosomes 1 and 5 were almost entirely syntenic with *M. truncatula* chromosomes 7 and 1, respectively. The remaining four chromosomes had large synteny blocks each with four or five *M. truncatula* chromosomes (Figure 1D). 

### 2.2. Genes Involved in PA Biosynthesis and Regulation

DMACA staining of organs of five legume species, namely *L. japonicus* MG-20, *M. truncatula*, *T. pretense*, *G. max*, and *P. vulgaris*, was performed, and PAs were found in the seed coat, stem, flower, and pod of *L. japonicus*. However, PAs were only found in the seed coat of *M. truncatula*, *G. max*, and *P. vulgaris* and in the flowers and seed coats of *T. pretense* (Figure 3A) [6]. 

Seventy-two PA biosynthesis-related genes were present in the *L. japonicus* genome, of which 62 were present in *M. truncatula*, 55 in *T. pretense*, 58 in *P. vulgaris*, 72 in *G. max*, and 29 in *Arabidopsis* (Appendix A). The copy numbers of the first key enzyme *PAL* and of genes in the late biosynthesis pathway encoding dihydroflavonol 4-reductase (*DFR*) and *TT2* in *L. japonicus* were significantly higher than those in the other species tested (Figure 3B). PAL catalyzes the conversion of L-phenylalanine to cinnamic acid, linking primary metabolism with secondary metabolism, a speed-limiting step in phenylpropanoid metabolism [7]. In total, 11 *PAL* genes were present in *L. japonicus* compared with 8 *PAL* genes in *G. max*, 7 in *T. pretense*, and 6 in *M. truncatula*. DFR is the first committed enzyme of the flavonoid pathway that leads to the production of common anthocyanins and PAs (Figure 4A). We identified six *DFRs* in the *L. japonicus* genome. A molecular phylogenetic tree of DFR from *L. japonicus*, *M. truncatula*, *A. thaliana*, *G. max*, *P. vulgaris*, and *T. pretense* was constructed using MEGA software 5.0 (Figure 4B). TT2 acts as a key determinant for PA accumulation. We identified three *TT2s* in the *L. japonicus* genome, whereas only one *TT2* was found in the other legumes tested (Figure 4C).

Furthermore, we examined the expression of PA biosynthesis-related genes from different tissues using the *Lotus Base* (https://lotus.au.dk/expat/, accessed on 23 December 2016). The expression of four genes, namely *Lj1gvBRI03262* (*F3′5′H*), *Lj1gvBRI09738* (*DFR*), *Lj1gvBRI01160* (*ANS*), and *Lj1gvBRI15634* (*ANR*), was downregulated in the roots and leaf but upregulated in the stem, petiole, flower, pod, and seed (Figure 3C). However, the aforementioned four genes in *M. truncatula* were downregulated in the root, stem, leaf, and flower and upregulated in the pod (Figure 4D). 

We propose that expansion of the PA biosynthesis-related genes, particularly the gene encoding the first key enzyme *PAL* and genes involved in the late biosynthesis pathway encoding *DFR* and *TT2* and transcription expression of *F3′5′H*, *DFR*, *ANS*, and *ANR* in most tissues, mainly contribute to PA accumulation in the stem, petiole, flower, pod, and seed coat of *L. japonicus*.

### 2.3. Transcriptome Analysis of the Response to Five Abiotic Stresses

We performed transcriptome analysis of responses to five abiotic stresses, namely exposure to 150 mM NaCl, 15% polyethylene glycol (PEG) 6000, 500 μM AlCl_3_ with pH 4.5, 37 °C and 0 °C for 6 h, with untreated seedings as the control. RNA-seq data analysis identified 2983, 2955, 6021, 4160, and 2740 differentially expressed genes (DEGs) under salt, PEG, heat (37 °C), low temperature/cold (0 °C), and AlCl_3_ (Appendix A), respectively (Figure 5A and Appendix A).

#### 2.3.1. Shared Responses under Abiotic Stresses

A total of 187 DEGs were common to all five stresses (Figure 5B and Appendix A). They were mainly enriched in steroid hormone biosynthesis, amino acid metabolism, homeostasis and transport, cell wall organization, and starch and sucrose metabolism (Figure 5C,D). *Lj1gvBRI23968.1*, *Lj1gvBRI35026.1*, and *Lj1gvBRI23372.1* were the top upregulated DEGs under the five stress conditions. Only nine TFs were common, including three *ERFs*, two *bZIPs*, one *MYB*, one *WRKY*, one *bHLH*, and one multiprotein bridging factor1c (*MBF*). Among them, the expression of only MBF (*Lj1gvBRI17497*) was upregulated and those of the other seven TFs were repressed in response to the five stresses.

Stress-specific transcription patterns are connected to upstream signaling via TFs [8]. We identified 548, 751, 2927, 1623, and 454 DEGs specifically in response to salt, PEG, heat, low temperature/cold, and AlCl_3_, respectively (Figure 5B). Among them, 48, 52, 110, 150, and 30 unique TF families were identified.

#### 2.3.2. Cold Stress-Specific Transcription Patterns

MG-20 is the most tolerant to low temperature among the 18 *L. japonicus* ecotypes [9]. We analyzed the specific DEGs under cold stress (Appendix A). Twenty-eight genes exhibited >100 fold change under cold stress (Figure 6A). Notably, two genes exhibited >1000 fold change in expression in cold stress, with the expression of *Lj1gvBRI02188* annotated as CCR4 (a serine/threonine protein kinase-like protein) being increased by 1022 fold. *Lj1gvBRI33915* annotated as unknown was the second top DEG. Twenty-five TF families (150 members) were exclusively expressed during cold stress, such as *AP2/ERF* (35), *C2C2* (18), *MYB* (13), *WRKY* (13), *bHLH* (10), and *GRAS* (9).

According to GO analysis results, 1633 cold stress-specific genes were significantly enriched in functions related to secondary metabolism, JA biosynthesis/metabolism process, and chlorophyll catabolic process. Meanwhile, the top 20 KEGG pathways included inflammatory mediator regulation of TRP channels, arginine and proline metabolism, and NOD-like receptor signaling pathway (Figure 6B,C and Appendix A).

#### 2.3.3. Heat Stress-Specific Transcription Patterns

We analyzed the specific DEGs under heat stress. Seven genes had a fold change of >10 under heat stress. Notably, increase in the expression of *Lj1gvBRI04909*, annotated as unknown, was the highest (by 11.6 folds) (Appendix A). *Lj1gvBRI17902* annotated as protein maintenance of meristems-like was the second top DEG (Figure 6D). A total of 15 TF families (110 members) were exclusively expressed during PEG stress, such as *AP2/ERF* (9), *FAR1* (8), *MYB* (7), *NAC* (7), *FHA* (6), and *HSF* (3).

Based on GO analysis results, 2927 heat stress-specific genes were significantly enriched in functions related to the glycolipid metabolic process, response to a topologically incorrect protein, and carbohydrate homeostasis. KEGG analysis indicated that these genes were enriched in the pathways related to cutin, suberine, and wax biosynthesis; starch and sucrose metabolism; and other pathways (Figure 6E,F and Appendix A). 

#### 2.3.4. Salt Stress-Specific Transcription Patterns

We analyzed the expression of specific DEGs under salt stress. The expression of three genes had a fold change of >5 under salt stress (Appendix A). Notably, the expression of *Lj1gvBRI14235*, annotated as the MYB TF, was increased 7.6-fold, indicating the highest increase (Figure 6G). In total, 17 TF families (47 members) were exclusively expressed during salt stress, such as *MYB* (9), *LOB* (6), *AP2/ERF* (5), and *bHLH* (5).

Based on the GO analysis results, 548 salt stress-specific genes were significantly enriched in functions related to the proline catabolic process, apocarotenoid metabolic process, anion transmembrane transport, and response to ionizing radiation. Meanwhile, according to KEGG analysis results, these genes were enriched in pathways related to steroid hormone biosynthesis, mineral absorption, and peroxisome. (Figure 6H,I and Appendix A).

#### 2.3.5. PEG Stress-Specific Transcription Patterns

We analyzed the expression of specific DEGs under drought stress. Three genes had a fold change of >4 under salt stress (Appendix A), Notably, expression of *Lj1gvBRI04219*, annotated as the arabinogalactan protein, increased by 6.3-fold, indicating the highest increase (Figure 7A). In total, 15 TF families (41 members) were exclusively expressed during PEG stress, such as *MYB* (8), *bHLH* (7), *NAC* (5), and *WRKY* (4).

Based on the GO analysis results, 752 drought stress-specific genes were significantly enriched in functions related to cell wall organization or biogenesis, regulation of stomatal complex development, retrograde transport, and response to SA. Meanwhile, the top 20 KEGG pathways were signaling pathways regulating stem cell pluripotency, and glycosphingolipid biosynthesis-lacto and neolact (Figure 7B,C and Appendix A).

#### 2.3.6. AlCl_3_ Stress-Specific Transcription Patterns

We analyzed the expression of specific DEGs under AlCl_3_ stress. Six genes had a fold change of >4 under AlCl_3_ stress (Appendix A). Notably, the expression of *Lj1gvBRI36827*, annotated as VQ domain-containing protein, was the highest by 5.7-fold. *Lj1gvBRI35647* annotated as casein kinase II subunit alpha was the second top DEG (Figure 7D). In total, 15 TF families (30 members) were exclusively expressed during AlCl_3_ stress, including *MYB* (6), *WRKY* (5), and *bHLH* (3).

According to GO analysis results, 454 AlCl_3_ stress-specific genes were significantly enriched in the fructose 6–phosphate metabolic process, response to aluminum ion, lipid modification, regulation of BR-mediated signaling pathway. Meanwhile, the top 20 KEGG pathways were related to folate biosynthesis, and drug metabolism–cytochrome P450 (Figure 7E,F and Appendix A). 

## 3. Discussion

Using comparative genomics and evolutionary analysis, this study provides insights into the important agricultural traits of *L. japonicus* including PA biosynthesis and response to abiotic stress. We revealed that the expansive and specific gene families of *L. japonicus* were enriched in secondary metabolite biosynthesis and the detection of external stimuli. Furthermore, according to the copy numbers and transcriptional pattern of PA biosynthesis-related genes, we proposed that the expansion of PAL, DFR, and TT2 in the PA biosynthesis pathway and the transcription expression of F3′5′H, DFR, ANS, and ANR in most tissues mainly contribute to PA accumulation in the stem, petiole, flower, pod, and seed coat of *L. japonicus*. Finally, we reported that MYB and AP2/ERF play crucial roles in response to abiotic stresses.

The genome of *L. corniculatus*, which accumulates PAs in whole plants, remains unknown. The relative model *L. japonicus* may provide clues for PA biosynthesis in most tissues and organs. PA accumulation is a complex quantitative trait, and 14 main enzymes are involved in PA biosynthesis from the aromatic amino acid. Here, 72 PA biosynthesis-related genes were identified in the *L. japonicus* genome, which is higher than those found in *M. truncatula* (62), *T. pretense* (55), *P. vulgaris* (58), and *Arabidopsis* (29) and the same as those in *G. max* (72). Especially, the expansion of PAL, which is the first speed-limiting key enzyme in phenylpropanoid biosynthesis that links primary metabolism with secondary metabolism [7]; DFR, which is the first committed enzyme of the flavonoid pathway that leads to anthocyanins and PAs [10]; and TT2s, which act as a key determinant for PA accumulation [11,12,13], have provided fundamental genetic materials for PA accumulation in more tissues. Furthermore, the expression of *F3′5′H*, *DFR*, *ANS* combined *ANR*, which provide the (-)-epicatechin extension, starter, and terminal units of PA polymers [14,15,16,17], were most strongly associated with PA accumulation in the stem, petiole, flower, pod, and seed coat of *L. japonicus*. In addition, expression profiles of 62 PA biosynthesis-related genes in *Medicago* showed that *ANS* and *ANR* were highly expressed only in the seed coat where the PAs accumulated exclusively (Figure 3C).

Drought, salinity, heat, cold, and AlCl_3_ stresses are considered major abiotic factors that negatively affect crop yield both qualitatively and quantitatively [4]. Identifying common and unique responses under these abiotic stresses is important for understanding the cross-talk mechanism [18]. According to the analysis of common transcriptional responses, the MBF gene *Lj1gvBRI17497.1* was screened. Its expression pattern was consistent with that of its homologue AtMBF1c in *Arabidopsis*, and its expression is specifically elevated in response to salinity, drought, heat, hydrogen peroxide, and pathogen infection. Furthermore, constitutive expression enhances the tolerance of transgenic plants to bacterial infection, heat, and osmotic stress [19]. Therefore, we propose that the gene MBF *Lj1gvBRI17497.1* may be an essential candidate gene having a critical role in response to multiple abiotic stresses.

Among the five types of abiotic stress, MG-20 is most tolerant to low temperature among the 18 *L. japonicus* ecotypes [9]. We found some clues based on the expression profile of specific DEGs under cold stress: there were more top DEGs (28 genes with fold change >100), and the expression level of the top DEGs was the highest (two genes with a fold change of >1000). Furthermore, these transcriptomic data and associated analysis results provide a resource for studying the response of *L. japonicus* to abiotic factors. According to shared and unique TF responses under abiotic stress, we proposed that MYB and AP2/ERF play more crucial roles under abiotic stresses.

In summary, based on our previous high-quality *L. japonicus* genome assembly and annotation, this study provides insights into the crucial agricultural traits of *L. japonicus*, including PA biosynthesis and response to abiotic stress by using comparative genomics and evolutionary analyses. We revealed the expansive and specific gene families of *L. japonicus* enriched in the biosynthesis of secondary metabolites and detection of external stimuli. We also proposed that expansion of PAL, DFR, and TT2 in the PA biosynthesis pathway and the transcription of *F3′5′H*, *DFR*, *ANS*, and *ANR* in most tissues mainly contribute to PA accumulation in the stem, petiole, flower, pod, and seed coat of *L. japonicus*. Lastly, we report that MYB and AP2/ERF play more crucial roles in abiotic stresses. This may provide valuable gene resources for legume forage abiotic stress resistance and nutrient improvement.

## 4. Materials and Methods

### 4.1. Phylogenetic Analysis and Gene Synteny Analysis

The genomes of *L. japonicus* (CNA0050696) and eight other plants, namely *M. truncatula*, *Glycine max*, T. pretense, *Phaseolus vulgaris*, C. arietinum, *Vigna unguiculata*, *Lupinus angustifolius*, and *Arabidopsis thaliana* (https://phytozome-next.jgi.doe.gov/, accessed on 22 November 2011), were used for evolutionary analysis. Paralogous and orthologous genes were identified using OrthoFinder [20]. The phylogenetic tree was constructed using RaxML [21]. The divergence times among the nine species were estimated using MCMCTREE within the PAML v4.9 package [22].

The expanded or contracted gene families were defined using CAFÉ v. 4 [23], and the significant genes were used for Kyoto Encyclopedia of Genes and Genomes (KEGG) pathway enrichment analysis. MCscanX was used to identify synteny blocks, and WGDI was used to determine the whole genome replication time (WGD) of *L. japonicus*, soybean, and alfalfa.

### 4.2. PA Staining with 4-Dimethylaminocinnamaldehyde

Healthy and fresh organs (roots, stems, leaves, flowers, and pods) of the reproductive stage were selected, decolorized in absolute ethanol containing 30% acetic acid for 12–18 h, stained with a cold 4-dimethylaminocinnamaldehyde (DMACA) reagent (0.3% *w*/*v* DMACA in 3N HCl/50% *w*/*v* ethanol), and washed three times using 75% ethanol. The stained organs can be protected in 70% ethanol, following which the color of PA-rich organs or tissues changes to blue [24]. The experiments were performed three times.

*Phaseolus vulgaris* (G19833), *Medicago truncatula* (A17), *Medicago sativa* (XinJiang DaYe), *Trifolium pratense* (Milvus B), *Glycine max* (Williams 82), Lotus japonicus (MG-20), *Trifolium repens* (Haifa), and *Arabidopsis thaliana* (Col-0) were used for visualization of PAs by DMACA staining. The plants were cultured at 25 ± 2 °C in a growth chamber with a photoperiod of 16 h/8 h.

### 4.3. Phylogenetic Tree and Transcriptional Expression Analysis

The complete amino acid sequences of proteins were aligned by ClustalX2 [25], and the phylogenetic tree was constructed by neighbor-joining algorithms of the MEGA 5.0 software (Test of phylogeny: Bootstrap method, No. of bootstrap replications: 1000, Model/Method: p-distance, Gaps/Missing data treatment: Pairwise deletion) [26].

The transcriptional expression data of *Medicago truncatula* were downloaded from the website https://mtsspdb.zhaolab.org/database/, accessed on 20 February 2020. We generated the heat maps of the expression patterns using TBtool II software [27].

### 4.4. Identification of Differentially Expressed Genes under Abiotic Stress

Total RNA was extracted using the RNAprep Pure plant Kit. mRNA was purified using oligo(dT)-linked magnetic beads following the manufacturer’s instructions. The purified mRNA was sequenced on the DNBSEQ platform, yielding 91.55 Mb data for each sample. RNA-seq reads were trimmed using SOAPnuke (version 1.5.6) [28] and mapped against *L. japonicus* reference genomes with Bowtie2 [29]. Gene expression values were calculated using the RSEM program [30]. Differential expression analysis was performed using DESeq2 [31]. GO and KEGG pathway enrichment analyses were performed using the phyper function in R package (*q*-value ≤ 0.05). The GO enrichment bubble chart was drawn using REVIGO (http://revigo.irb.hr/), then the exported R language pack was run by Rstudio.

### 4.5. Plant Materials and Treatments

*L. japonicus* seeds were treated with 96% sulfuric acid (H_2_SO_4_) for 10 min and washed five to six times in sterile distilled water. Subsequently, seeds were placed on the Murashige–Skoog (MS) agar medium in a growth chamber with 16 h of light/8 h of dark; 30-day-old seedlings were transferred to the MS liquid medium supplemented with different stress solutions (150 mM NaCl, 15% polyethylene glycol 6000, 500 µM AlCl_3_ with pH 4.5, 37 °C, 0 °C), MS medium as the control. The seedlings for RNA-seq were harvested 6 h after treatment; untreated seedlings were used as control. All samples were frozen in liquid nitrogen and stored at −80 °C until use.

## 5. Conclusions

In summary, based on our previous high-quality *L. japonicus* genome assembly and annotation, this study provides insights into the crucial agricultural traits of *L. japonicus*, including PA biosynthesis, and response to abiotic stress by using comparative genomics and evolutionary analyses. We revealed the expansive and specific gene families of *L. japonicus* enriched in the biosynthesis of secondary metabolites and detection of external stimuli. We suggested that increased copy numbers and transcription of PA-related genes contribute to PA accumulation in the stem, petiole, flower, pod, and seed coat of *L. japonicus*. Meanwhile, according to shared and unique transcription factors responding to five abiotic stresses, we revealed that MYB and AP2/ERF play more crucial roles in abiotic stresses. Our study provides new insights into the key agricultural traits of *L. japonicus* including PA biosynthesis and response to abiotic stress. This may provide valuable gene resources for legume forage abiotic stress resistance and nutrient improvement.

## Figures and Tables

**Figure 1 plants-13-01151-f001:**
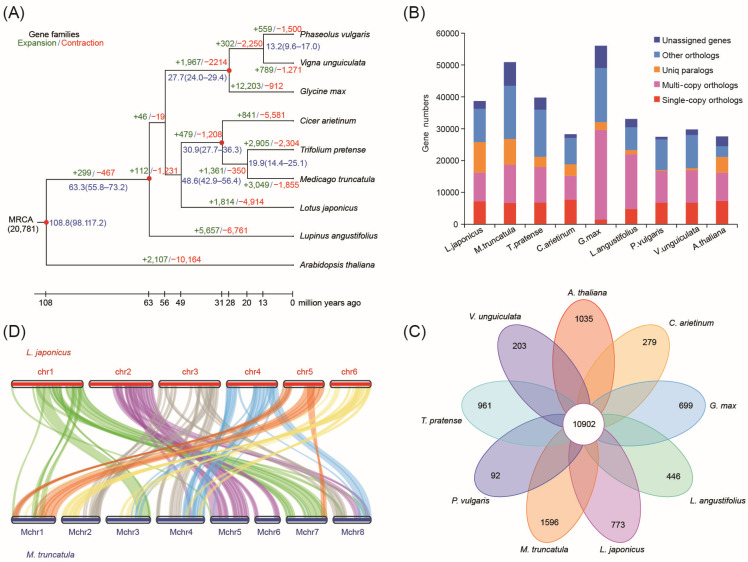
Evolution and synteny of the Lotus japonicus genome. (**A**) The phylogenetic relationship and split-time estimation are based on all single-copy gene families shared by all species used in this analysis. (**B**) An overview of ortholog and paralog genes among *L. japonicus* and eight other plant species (Mt, *Medicago truncatula*, Gm, *Glycine max*, Tp, *Trifolium pretense*, Pv, *Phaseolus vulgaris*, Ca, *Cicer arietinum*, Vu, *Vigna unguiculata*, La, *Lupinus angustifolius*, and Ath, *Arabidopsis thaliana*). (**C**) Venn diagram showing the shared and unique gene families among *L. japonicus* and eight other plant species. (**D**) Synteny analysis between *L. japonicus* and *M. truncatula*.

**Figure 2 plants-13-01151-f002:**
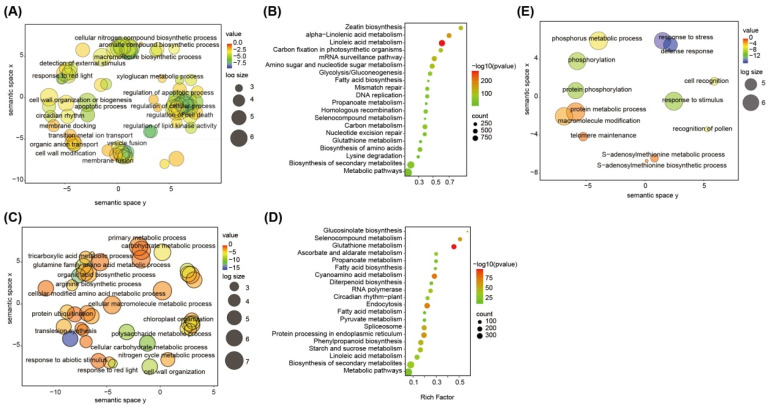
Functional enrichment of the *L. japonicus*-specific gene families and expansion or contraction of gene families. (**A**) GO enrichment analysis of specific gene families. (**B**) Statistics for the top 20 enriched pathways among the specific gene families. The degree of KEGG enrichment was determined by the enrichment factor, *q*-value, and gene number. The sizes and colors of spots represent the number of genes and the *q*-value. (**C**) GO enrichment analysis of contraction gene families. (**D**) GO enrichment analysis of expansion gene families. (**E**) Bubble plot of KEGG of expansion gene families.

**Figure 3 plants-13-01151-f003:**
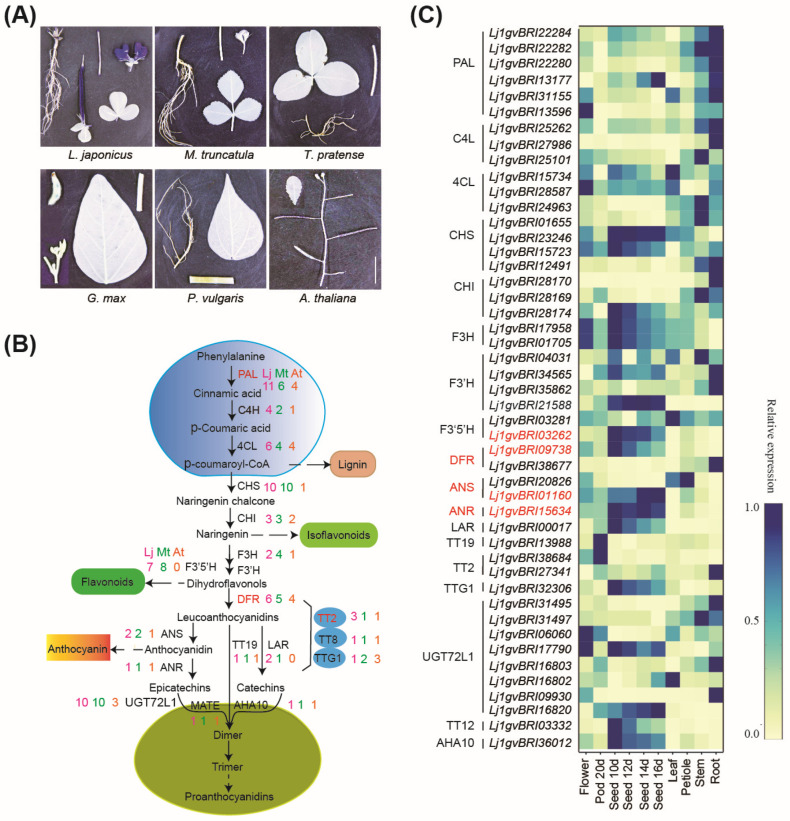
Analysis of PA biosynthesis-related genes. (**A**) DMACA staining of different tissues from five legume species, including *L. japonicus* MG-20, *M. truncatula*, *T. pretense*, *G. max*, and *P. vulgaris*, and a nonlegume species *A. thaliana*. (**B**) Diagram of the PA biosynthesis pathway and copy numbers of genes involved in the PA biosynthesis pathway in *L. japonicus*. *PAL*: Phenylalanine ammonia lyase, *C4H*: cinnamate 4-hydroxylase, *4CL*: 4-coumarate coenzyme A ligase, *CHS*: chalcone synthase, *CHI*: chalcone isomerase, *F3H*: flavonoid 3-hydroxylase, *F3′H*: flavanone’-hydroxylase, *F3′5′H*: flavonoid 3,5-hydroxylase, *DFR*: dihydroflavonol 4-reductase, *LAR*: leucocyanidin reductase, *ANS*/*LDOX*: anthocyanidin synthase/leucoanthocyanidin dioxygenase, *ANR*: anthocyanidin reductase, *GST* (*TT19*): glutathione S-transferase, *TT2*: transparent testa 2, *TT8*: transparent testa 8, *TTG1*: transparent testa glabra 1, *MATE* (*TT12*): multidrug and toxic compound extrusion protein, *UGT72L1*: epicatechin glucosyltransferase, and *AHA10*: *Arabidopsis* H+-ATPase 10. (**C**) Heatmaps showing the expression profiles of genes involved in PA biosynthesis pathways in different organs from *L. japonicus*. The expression values of genes at the row scale were normalized, and the values are indicated by a continuous color scheme. Blue indicates a high expression, and yellow indicates a low expression.

**Figure 4 plants-13-01151-f004:**
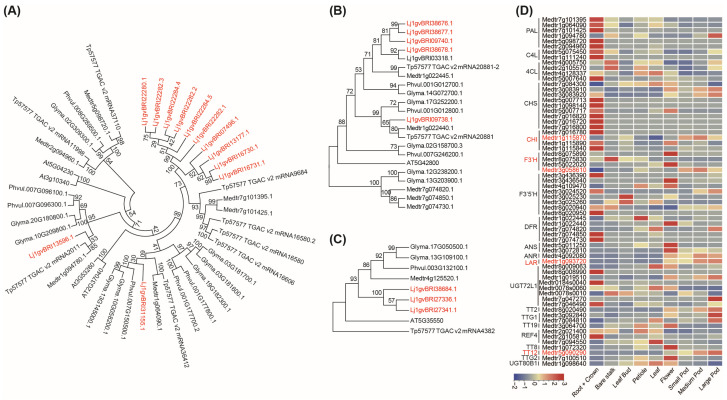
Transcription expression analysis and phylogenetic tree of proanthocyanidins-related genes. (**A**) Phylogenetic relationships of PAL from *L. japonicus*, *M. truncatula*, *T. pretense*, *P. vulgaris*, *G. max* and *A. thaliana*. (**B**) Phylogenetic relationships of DFR. (**C**) Phylogenetic relationships of TT2. (**D**) Heatmaps showing expression profiles of the genes involved in proanthocyanidin biosynthesis pathways in the different organs from *M. truncatula*.

**Figure 5 plants-13-01151-f005:**
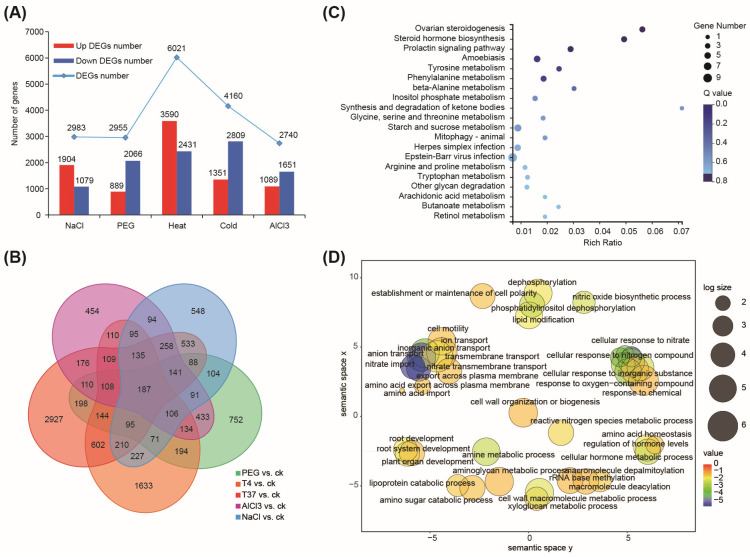
Differential gene expression analysis of response to abiotic stressors. (**A**) Number of genes expressed differentially in response to salt, drought, heat, cold, and AlCl_3_ stresses. (**B**) Venn diagram showing the shared and unique DEGs in response to abiotic stresses. (**C**) Bubble plot of KEGG of 187 DEGs common to all five stresses. (**D**) GO enrichment analysis of 187 DEGs common to all five stresses.

**Figure 6 plants-13-01151-f006:**
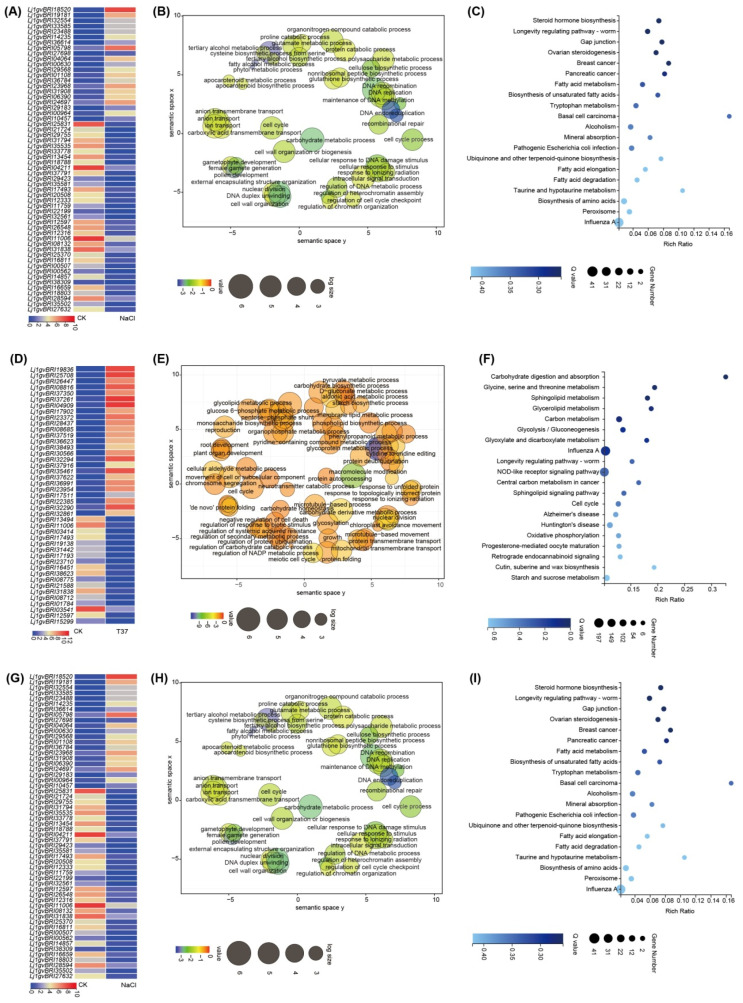
Cold, heat and salt stress-specific transcription patterns. (**A**) Heatmaps showing expression profiles of top 38 specific DEGs under cold stress. (**B**) GO enrichment analysis of 1633 specific DEGs under cold stress. (**C**) Bubble Plot of KEGG of 1633 specific DEGs under cold stress. (**D**–**F**) Heat stress-specific transcription patterns. (**G**–**I**) Salt stress-specific transcription patterns.

**Figure 7 plants-13-01151-f007:**
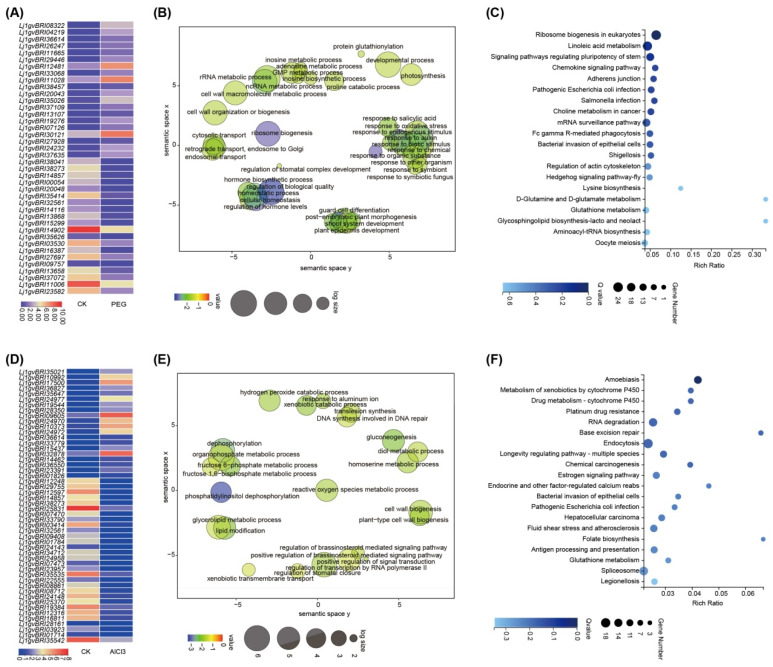
Drought and AlCl_3_ stress-specific transcription patterns. (**A**) Heatmaps showing expression profiles of top 40 specific DEGs under salt stress. (**B**) GO enrichment analysis of 752 specific DEGs under salt stress. (**C**) Bubble Plot of KEGG of 752 specific DEGs under salt stress. (**D**–**F**) AlCl_3_ Stress-specific transcription patterns.

## Data Availability

All data generated or analyzed during this study are included in this published article (and its Appendix A Files). All RNA sequencing data in this study are available on SRA through accession numbers SRP399738 (Cold stress), SRP399738 (Heat stress), SRP399738 (NaCl stress), SRP399738 (PEG stress), SRP399737 (AlCl_3_ stress), and SRP399738 (CK).

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
