# Peer review of "Comparative Genomics of Lotus japonicus Reveals Insights into Proanthocyanidin Accumulation and Abiotic Stress Response"

_plants, 2024, doi:10.3390/plants13081151_

Round 1
Reviewer 1 Report
Comments and Suggestions for Authors
The manuscript by Sun Zhanmin and colleagues describe a study where the impact of different abiotic stress conditions on proanthocyanidin biosynthesis of Lotus japonicus.
The topic studied is relevant, because until now, there are not many studies on the biosynthesis of proanthocyanidins. Knowledge of the biosynthetic process of these compounds still has some gaps. These are even greater in the case of possible impacts from various abiotic factors.
Despite the relevance of this study, there are some doubts that the authors should clarify in the article.
1 - At the end of the introduction, authors should be clearer about the aim of the work and their relevance.
2 - Line 187-188: It is not clear how the tests for the impact of abiotic factors were established. There are details that must be described in the text. It will be useful to the readers. This comment should be considerate in the following items: 2.3.1.; 2.3.2; 2.3.3.; 2.3.4.; 2.3.5.
Figure 6, it’s very complex, particularly for (B), (E) and (H).
3 - It should be clear and well defined when the study was carried out in terms of the vegetative development of the plant. At what time ? It is known that the dynamics of proanthocyanidins biosynthesis varies throughout plant development.
4 - It would be appropriate for the authors to present some quantification of proanthocyanidins. Therefore, it would be interesting to analyze the impact of abiotic factors on biosynthesis and consequently on the contents present in the plant tissues analyzed.
Author Response
1 - At the end of the introduction, authors should be clearer about the aim of the work and their relevance.
Reply:Agree. At the end of the introduction, “This may provide valuable gene resources for legume forage abiotic stress resistance and nutrient improvement.” Were added.
2 - Line 187-188: It is not clear how the tests for the impact of abiotic factors were established. There are details that must be described in the text. It will be useful to the readers. This comment should be considerate in the following items: 2.3.1.; 2.3.2; 2.3.3.; 2.3.4.; 2.3.5.
Reply: Agree. Line158-162 “We performed transcriptome analysis of responses to five abiotic stresses, namely exposure to salt, drought, heat, cold, and AlCl3. Through RNA-seq data (Table S8–10), 2983, 2955, 6021, 4160, and 2740 genes differentially expressed (DEGs) were screened in response to salt, polyethylene glycol (PEG), heat (37℃), low temperature/cold (0℃), and AlCl3, respectively (Figure 5A; Table S11).”were revised as “We performed transcriptome analysis of responses to five abiotic stresses, namely exposure to 150 mM NaCl, 15% polyethylene glycol 6000, 500 μM AlCl3 with pH 4.5, 37 ℃ and 0℃ for 6 hours, untreated seedings as the control. Through RNA-seq data (Table S8–10), 2983, 2955, 6021, 4160, and 2740 genes differentially expressed (DEGs) were screened in response to salt, polyethylene glycol (PEG), heat (37℃), low temperature/cold (0℃) and AlCl3, respectively (Figure 5A; Table S11).”
Figure 6, it’s very complex, particularly for (B), (E) and (H).
Reply: We identified 548, 751, 2927, 1623, and 454 DEGs specifically in response to salt, PEG, heat, low temperature/cold, and AlCl3, respectively, To highlight prominent DEGs, in the figure 6, only top 38-40 genes were showed in heatmaps (A), top 40-50 GO enrichment analysis data were showed (B,E,H).
3 - It should be clear and well defined when the study was carried out in terms of the vegetative development of the plant. At what time ? It is known that the dynamics of proanthocyanidins biosynthesis varies throughout plant development.
Reply: Agree. line 330-331, “Healthy and fresh organs (roots, stems, leaves, flowers, and pods) were selected” were revised as “Healthy and fresh organs (roots, stems, leaves, flowers, and pods) of reproductive stage were selected”.
4 - It would be appropriate for the authors to present some quantification of proanthocyanidins. Therefore, it would be interesting to analyze the impact of abiotic factors on biosynthesis and consequently on the contents present in the plant tissues analyzed.
Reply: This is a good idea, in our study, Considering that alfalfa (Medicago sativa) and white clover (Trifolium repens) can not accumulate PAs in the leaf and stem tissues, not the problem of low content, thus mainly to solve problems with or without, nextly, we will analyze the impact of abiotic factors on biosynthesis and consequently on the contents present in the plant tissues analyzed.

Reviewer 2 Report
Comments and Suggestions for Authors
The authors of the submitted manuscript compared the genome of L. japonicus with the genomes of other legume species. They found that the specific and expanded L. japonicus gene families were enriched in secondary metabolism biosynthetic genes. Authors suggested that increased copy numbers of proanthocyanidin-related genes contribute to PA accumulation in the stem, petiole, flower, pod, and seed coat of L. japonicus. They also revealed that MYB 22 and AP2/ERF are critical in abiotic stress responses. The study provides insights into PA biosynthesis and response to abiotic stress in L. japonicus.
The manuscript will benefit from the suggestions below:
1. Figure 3A. I suggest using a different background than blue for the figure. The blue background masked the DMACA staining.
2. Figure 3C. Provide color code of expression.
3. Rephrase the sentence to make it clear: "Through RNA-seq data (Table S8–10), 2983, 2955, 6021, 4160, and 2740 genes differentially expressed (DEGs) were screened in response to salt, polyethylene glycol (PEG), heat (37℃), low temperature/cold (0℃), and AlCl3, respectively". Do you mean that the listed numbers of genes were identified as DEGS in response to different stresses?
4. Please describe methods for Gene Ontology and visualization of bubble plots.
Comments on the Quality of English LanguageThe manuscript will benefit from improving the English language.
Author Response
- Figure 3A. I suggest using a different background than blue for the figure. The blue background masked the DMACA staining.
Reply: Agree. The color of background is revised to be gray.
- Figure 3C. Provide color code of expression.
Reply: Agree. color code of expression was added in the figure 3C.
- Rephrase the sentence to make it clear: "Through RNA-seq data (Table S8–10), 2983, 2955, 6021, 4160, and 2740 genes differentially expressed (DEGs) were screened in response to salt, polyethylene glycol (PEG), heat (37℃), low temperature/cold (0℃), and AlCl3, respectively". Do you mean that the listed numbers of genes were identified as DEGS in response to different stresses?
Reply: Agree. the listed numbers of genes were identified as DEGS in response to different stresses. The sentence “Through RNA-seq data (Table S8–10), 2983, 2955, 6021, 4160, and 2740 genes differentially expressed (DEGs) were screened in response to salt, polyethylene glycol (PEG), heat (37℃), low temperature/cold (0℃), and AlCl3, respectively” were revised as “RNA-seq data analysis identified 2983, 2955, 6021, 4160, and 2740 differentially expressed genes (DEGs) under salt, polyethylene glycol (PEG), heat (37℃), low temperature/cold (0℃), and AlCl3 (Table S8–10), respectively.”
- Please describe methods for Gene Ontology and visualization of bubble plots.
Reply: line 423, “then the exported R language pack was ran by Rstudio” were added.
